

# Effectiveness of small road tunnels and fences in reducing amphibian roadkill and barrier effects at retrofitted roads in Sweden

Jan Olof Helldin[1],* and Silviu O. Petrovan[2,3,*]

[1] Swedish Biodiversity Centre, Swedish University of Agricultural Sciences, Uppsala, Sweden
[2] Department of Zoology, University of Cambridge, Cambridge, UK
[3] Froglife, Peterborough, UK
* These authors contributed equally to this work.

## ABSTRACT

Schemes to reduce road impacts on amphibians have been implemented for decades in Europe, yet, several aspects on the effectiveness of such schemes remain poorly understood. Particularly in northern Europe, including Sweden, there is a lack of available information on road mitigation for amphibians, which is hampering implementation progress and cost-effectiveness analyses of mitigation options. Here, we present data derived from systematic counts of amphibians during spring migration at three previous hot-spots for amphibian roadkill in Sweden, where amphibian tunnels with guiding fences have been installed. We used the data in combination with a risk model to estimate the number of roadkills and successful crossings before vs. after mitigation and mitigated vs. adjacent non-mitigated road sections. In mitigated road sections, the estimated number of amphibians killed or at risk of being killed by car traffic decreased by 85–100% and the estimated number successfully crossing the road increased by 25–340%. Data, however, suggested fence-end effects that may moderate the reduction in roadkill. We discuss possible explanations for the observed differences between sites and construction types, and implications for amphibian conservation. We show how effectiveness estimates can be used for prioritizing amphibian passages along the existing road network. Finally, we emphasize the importance of careful monitoring of amphibian roadkill and successful crossings before and after amphibian passages are constructed.

## INTRODUCTION

Amphibian populations may be severely impacted by road mortality and barrier effects of roads and traffic (*Hels & Buchwald, 2001*; *Gibbs & Shriver, 2005*; *Jaeger & Fahrig, 2004*; *Nyström et al., 2007*; *Beebee, 2013*). Mass mortalities of amphibians often occur where roads cut across annual migration routes between hibernation and breeding habitats. Roadkill, habitat loss and the generally harsh environment for amphibians along roads can

Corresponding author
Jan Olof Helldin, j-o.helldin@slu.se

also lead to avoidance and barrier effects (*De Maynadier & Hunter, 2000*; *Fahrig & Rytwinski, 2009*), preventing them from reaching crucial habitats or resources. Aiming to reduce such negative effects, road mitigation measures have been developed and implemented for over 40 years in Europe (*Langton, 2015*). However, monitoring of mitigation measures is often lacking or insufficient (e.g., focusing solely on usage) and previous studies have shown varying results (*Brehm, 1989*; *Meinig, 1989*; *Zuiderwijk, 1989*; *Puky & Vogel, 2003*; *Mechura et al., 2012*; *Faggyas & Puky, 2012*; *Ottburg & Van Der Grift, 2019*; *Matos et al., 2019*). Consequently, numerous aspects on the actual effectiveness of road mitigation schemes for amphibians remain poorly understood, holding back planning efforts and opportunities for improvements.

Well-functioning mitigating schemes for amphibians are strongly needed as populations of amphibians continue to decline in Europe, including some of the main target species for road mitigation, the common toad (*Bufo bufo*), the common frog (*Rana temporaria*) and the great crested newt (*Triturus cristatus*) (*Bonardi et al., 2011*; *Beebee, 2013*; *Petrovan & Schmidt, 2016*; *Kyek, Kaufmann & Lindner, 2017*). However, in northern Europe, including Sweden, there is a widespread lack of available information on the effectiveness of road mitigation for amphibians. This is particularly concerning due to the well-developed road network and the potentially complex effects of the harsher climate on microclimatic conditions inside wildlife underpasses or other unforeseen aspects. The absence of structured information and evidence of effectiveness is hampering implementation progress and much needed cost-effectiveness analyses of mitigation options.

To minimize the road impacts on amphibians, road managers in and near Stockholm (the Swedish Transport Administration and Stockholm Municipality) constructed passages for amphibians at three sites where large concentrations of amphibians were killed on roads, particularly during spring migration, and thus were considered road sections in critical need of ecological mitigation. The passages were in the form of permanent tunnels with double-sided guiding fences intended to lead the amphibians safely under the road in both directions. The constructions largely followed the European (*Iuell et al., 2003*) and Swedish (*Eriksson, Sjölund & Andren, 2000*; *Banverket, 2005*) guidelines for design and dimensions, however with tunnels narrower than the recommended minimum diameter 0.6–1 m and with a distance between neighboring tunnels in some cases longer than the recommended maximum of 30–60 m.

Before and after the construction of these passages, the number and location of amphibians on the road as well as along the fences and in the tunnels were recorded, as the basis for planning of the mitigation constructions and monitoring of their effectiveness. Here, we summarize the results of these counts, and discuss the implications in terms of reduced roadkill and barrier effect, differences between constructions and improved amphibian conservation. We propose a baseline for prioritizing amphibian passages along the existing road network, and suggest some directions for further studies that would support the planning of amphibian mitigation schemes.

**Table 1 Characteristics of the roads and the amphibian mitigation measures at the three study sites near Stockholm, Sweden.**

| Site | 1. Skårby | 2. Kyrksjölöten | 3. Skeppdalsström |
|---|---|---|---|
| Location | 59°13′34N 17°43′55E | 59°20′53N 17°55′35E | 59°18′16N 18°29′32E |
| Construction year of mitigation measure | 2005, additional tunnels in 2008 | 2014 | 2015 |
| Road | | | |
|   Name/no | Road 584 | Spångavägen | Road 222 |
|   Owner/manager | Swedish Transport Administration | Stockholm Municipality | Swedish Transport Administration |
|   Mitigated section (m) | 300 | 315 | 190 + 110 |
|   Traffic (daily average)[a] | 3,000 | 7,800 | 8,600 |
|   Width (m) | 7 | 16[b] | 7 |
| Guiding fences (barriers) | | | |
|   Height | 40 | 45 | 40 |
|   Material | Cement concrete | Polymer concrete | Metal |
|   Sides | Double sided | Double sided | Double sided |
|   Location | Parallel to road | Parallel to road | Parallel to road |
|   End | Wide V-shape | U-shape | Narrow U-shape |
|   Top | Straight | Angled | Angled |
| Tunnels | | | |
|   Type | Closed top circular | Closed top dome | Closed top circular |
|   Guiding structure | (T-shape with roof)[c] | I-shape | None |
|   Number | 5 | 2 | 5 |
|   Diameter (cm) | 40  50  40  40  40 | 50 × 32 (both) | 30 (all) |
|   Length (m) | 11  ?  11  16  12 | 25      19 | 10 (all) |
|   Material[d] | M  Cc  M  M  M | Pc      Pc | P  P  P  M  P |
|   Water[e] | R  R  D  R  R | S      R | D  S  S  D  R |
|   Max water depth (cm) | 10  5  –  5  5 | 5      1 | –  30  25  –  5 |
|   Distance between (m) | 55  55  70  75 | 180 | 47    55  215[f]  115 |

**Notes:**

Data on individual tunnels are listed from east to west (see Fig. 2).
[a] Data from 2007 to 2015.
[b] Including pedestrian and bike lanes.
[c] Not clear whether these were in place during monitoring.
[d] M, metal; Cc, cement concrete; Pc, polymer concrete; P, plastic.
[e] R, running; D, dry; S, standing (at the time for fieldwork).
[f] Including distance between mitigated sections.

# MATERIAL AND METHODS

## Study sites and available field data

The three monitored sites are similar in several respects. The roads are all of intermediate size (seven to eight m wide, ca 3,000–9,000 vehicles per average day; Table 1), and mainly used for local and commuting traffic in Stockholm metropolitan area (Fig. 1). The landscape is a small-scale valley terrain at 10–30 m elevation, with a mix of forest, farmland and housing/garden areas. The mitigated road sections all have an important amphibian breeding wetland of around 5–10 ha nearby (Fig. 2) and main overwintering

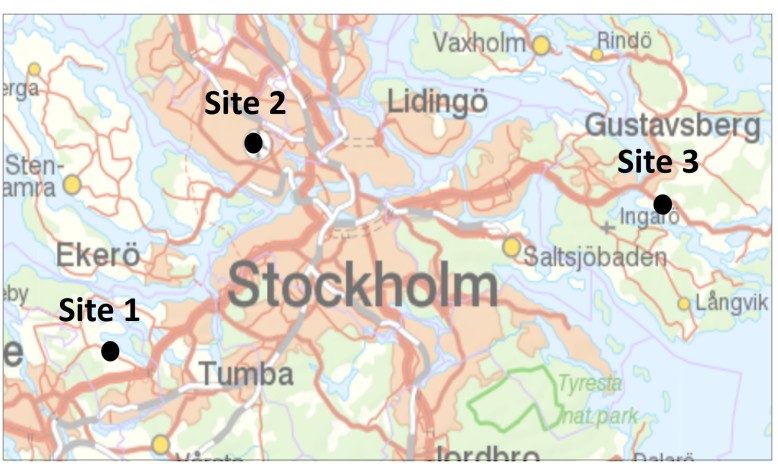

**Figure 1 Location of the three study sites in Stockholms larger metropolitan area.** Map image credit: Lantmäteriet.

habitat, typically woodland, on the opposite side of the road). Before mitigation, the road sections were well known hot-spots for amphibian roadkill during spring migration. The amphibian species diversity in the region is limited, with only five species occurring; common toad, common frog, moor frog (*Rana arvalis*), smooth newt (*Lissotriton vulgaris*) and great crested newt.

The mitigation systems are roughly similar in terms of dimensions of tunnels and fences and length of road section mitigated, although there are some differences in exact dimensions and material of the constructions (Table 1; Fig. 3). At all sites, tunnels were impacted by running or standing water to a varying degree during the studies (Table 1).

Live and dead amphibians were counted along the road prior to construction of the passage ("before"), aiming to identify the most critical road sections for mitigation and to locate major migration routes where tunnels should be placed. Amphibians were also counted post-mitigation ("after"), along the road, along fences and in tunnels, to assess the anticipated reduction in roadkill and evaluate the use of the tunnels. While the field efforts varied between sites and periods (Table 2 and site descriptions below), all data collection was conducted during peak spring migration, with methods that could be considered comparable in terms of number of amphibians found per time and road interval.

### Site 1 Skårby

The pond and wetland at Skårby has one of Sweden's largest breeding populations of great crested newt (>300 individuals) and also a large breeding population of smooth newt (>2,000 individuals; *Peterson & Collinder, 2006*). The amphibian mitigation system was constructed in phases; 300 m permanent fence with three tunnels was constructed in 2005 and two additional tunnels were constructed in 2008. Amphibians on the road were counted in one night in the year before mitigation (2004), and in four nights with the mitigation in place (2008). The road section searched was ca 520 m, extending in both directions ≥150 m outside of the section to be mitigated. Live animals and fresh carcasses

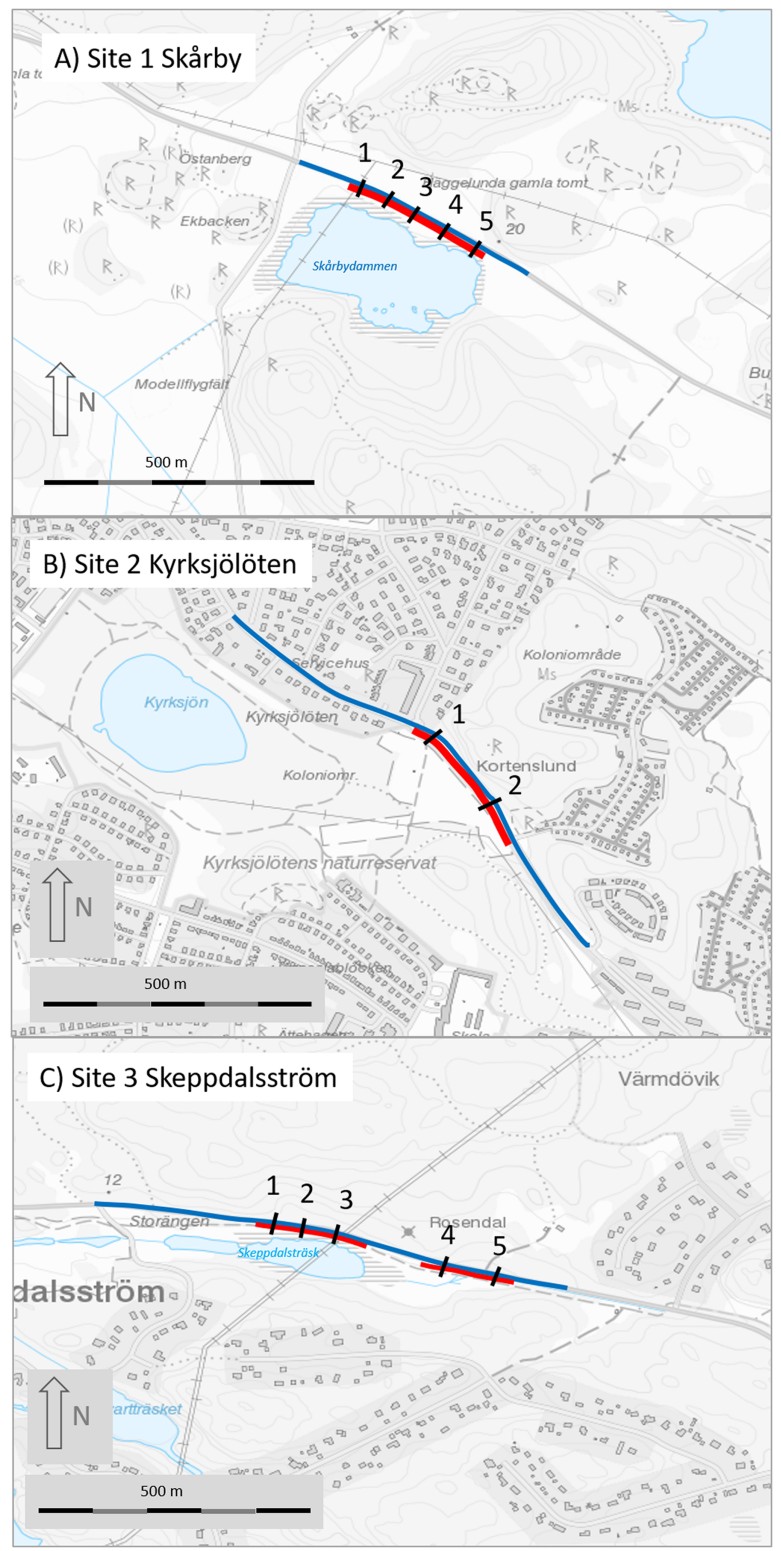

**Figure 2 Maps of the three study sites (A, site 1 Skårby; B, site 2 Kyrksjölöten; C, site 3 Skeppdalsström).** Red lines denote mitigated (fenced) section, black lines are the tunnels and blue line is the road section where amphibians were counted before and after mitigation. Map image credit: Lantmäteriet.

### Site 1 Skårby

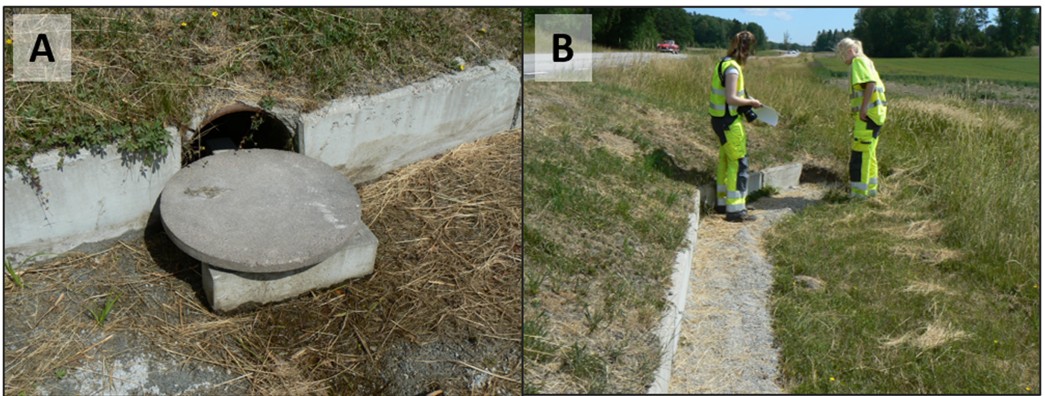

### Site 2 Kyrksjölöten

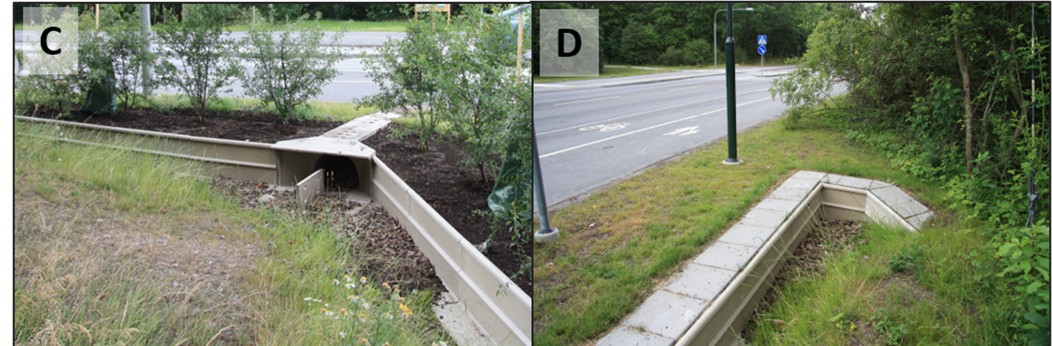

### Site 3 Skeppdalsström

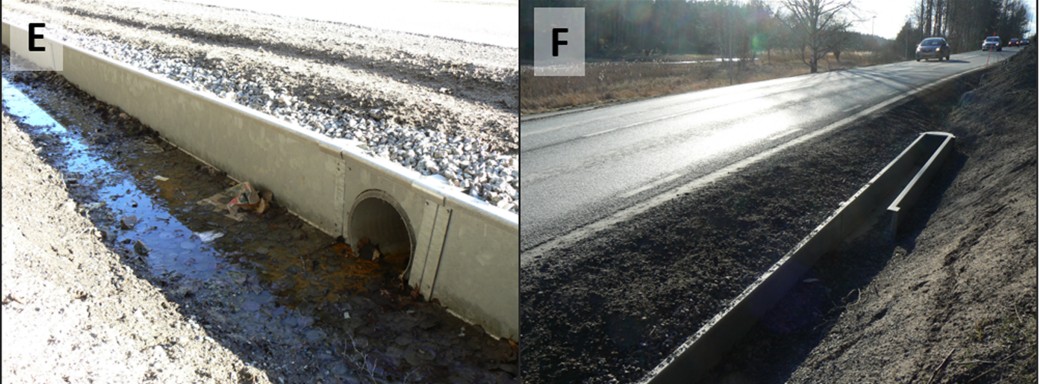

**Figure 3 Amphibian tunnel with guiding structure, fence and fence-end at the three study sites (A–B) site 1 Skårby; (C–D) site 2 Kyrksjölöten; (E–F) site 3 Skeppdalsström.** Photos: Jan Olof Helldin and Erik Jondelius.

(from the current night) were counted. Trapping in tunnels was conducted during five nights in total (two in 2010 and three in 2013). Bow net traps were mounted on the tunnel exits (i.e., the opening on the wetland side) to count amphibians passing through the tunnels toward the wetland. One of the tunnels (no. 2) could not be monitored because the exit was completely under water; however this tunnel was in place already before the mitigation system was constructed, functioning as a drainage pipe, and it was therefore not

**Table 2 Amphibian data collection methods and efforts at the three study sites near Stockholm, Sweden.**

| Site | 1. Skårby | | 2. Kyrksjölöten | | 3. Skeppdalsström | |
|---|---|---|---|---|---|---|
| | Before | After | Before | After | Before | After |
| Visual search | | | | | | |
|   Section searched (m) | 520 | | ca 1,000 | | ca 950 | |
|   No. of nights | 1 | 4 | 17[a] | 3[b] | 7 | 4 |
|   Time period | April 15–16, 2004 | April 6–22, 2008 | March 27–May 9, 2012 | April 8–15, 2015 | April 7–19, 2015 | April 7–18, 2016 |
| Pitfall trapping along temporary fences | | | | | | |
|   Section trapped (m) | – | | 350 | – | – | |
|   No. of nights | – | | 17[a] | – | – | |
|   Time period | – | | March 27–May 9, 2012 | – | – | |
| Net trapping | | | | | | |
|   No. of tunnels | – | 4 | – | | – | |
|   No. of nights | – | 5 | – | | – | |
|   Time period | – | April 9–11, 2010, April 15–18, 2013 | – | | – | |
| Camera trapping | | | | | | |
|   No. of tunnels | – | | – | 2 | – | 4 |
|   No. of nights | – | | – | 32[c] | – | 7–11[d] |
|   Time period | – | | – | April 1–May 3, 2015 | – | April 5–23, 2016 |

Notes:
[a] Representing seven significant migration nights.
[b] Representing a period of 8 days and nights.
[c] Representing 14 significant migration nights.
[d] Differed between tunnels; see Table 4.

further considered in the analyses. All study nights were selected to represent important migration nights (suitable weather conditions and timing). Position, species, status (e.g., dead/alive) and time was recorded for all amphibian observations, both on the road and in the tunnels. Due to the dominance of newts at this site (ca 98% of amphibians observed) we excluded data on other species, and we pooled the data on the two newt species in the analyses. Most of the newts found when searching the road were dead (ca 72%).

### Site 2 Kyrksjölöten

The lake Kyrksjön including adjacent wetlands in the nature reserve Kyrksjölöten has a large breeding populations of common toad (the exact number has however not been assessed). The numbers of other amphibians are small. The amphibian mitigation system was constructed at the major road (Spångavägen) going past the area, in connection to an upgrade of the road in autumn 2014. Amphibians were counted during 17 evenings in 2012, before mitigation was installed, at a one-sided temporary fence and pitfall traps along the section to be permanently mitigated, and by searching the road and verges. Only on 7 of the 17 nights a relatively large number of amphibians were found or trapped, and

accordingly could be labeled significant migration night. Amphibians were counted during three evenings in 2015, with the mitigation in place, along the permanent fences and on the road and verges. Evenings for fieldwork were selected to represent important migration evenings (suitable weather conditions and timing). Dead amphibians had accumulated between evenings, thus representing a total period of ca 8 days. The total road section searched was ca 1,000 m (same section before and after), therefore extending in both directions >200 m outside of the fenced section. Customized infrared timelapse cameras (15 s interval) assembled by Froglife (Peterborough, UK; *Jarvis, Hartup & Petrovan, 2019*) were mounted on the ceiling inside both tunnel entrances during 32 days in 2015. Only on 14 of the 32 days a significant number of amphibians were recorded, and accordingly could be labeled significant migration night. Position, species, status (e.g., dead/alive) and time were recorded for all amphibian observations, both on the road, along fences and in the tunnels. For animals on tunnel photos, movement direction (in or out) was noted and the minimum net number through the tunnels was calculated. Due to the dominance of common toads at this site (ca 99% of amphibians observed) we excluded data on other species. Most of the toads found when searching the road were dead (ca 82%), while all toads found or captured along the temporary fence were alive.

### Site 3 Skeppdalsström

The wetland Skeppdalsträsk serves as breeding area for all five amphibian species. Breeding populations during studies were estimated to 600 common toads, 150 common frogs and 60 moor frogs (*Andersson & Lundberg, 2015*); smooth newt was not included in the assessment but is probably at similar abundance with common toads, while great crested newt was not discovered until 2017 (A Crussell, 2017, personal communication). Volunteers have been active on the site since 2013, moving amphibians across the road during spring migration. The amphibian mitigation system was constructed in summer 2015. Amphibians were counted during seven evenings in 2015, before mitigation was installed, by searching the road, including the verge on the northern side, and during four evenings in 2016, with the mitigation in place, along the permanent fences and on the road and northern verge. Evenings for fieldwork were selected to represent important migration evenings (suitable weather conditions and timing). Each evening, at least five people took part in the search, regularly patrolling the road, and accordingly most amphibians were found alive before or when entering the road. The road section searched was ca 950 m (same section before and after), therefore extending between and in both directions ≥100 m outside of the mitigated sections. Customized infrared timelapse cameras (as for site 2) were mounted on the ceiling inside of the tunnel entrances; due to temporary failure of the IR light source, the total number of camera days acquired varied between 7 and 11 (Table 3). One of the tunnels (no. 5) was not monitored because of a constant flow of water inside the tunnel, which was assumed to interfere with the analysis of tunnel photos; however this tunnel was in place already before the mitigation system was constructed, functioning as a drainage pipe, and it was therefore not further considered in the analyses. Position, species, status (e.g., dead/alive) and time were
**Table 3 Estimated number of amphibians killed per night along the studied road sections before and after mitigation, separated between mitigated and adjacent non-mitigated sections.**

| Section | Before | After | Δ |
|---|---|---|---|
| **Site 1. Skårby** | | | |
| Mitigated | 228 | 10 | −218 |
| Non-mitigated | 91 | 60 | −31 |
| Total | 319 | 70 | −249 |
| **Site 2. Kyrksjölöten** | | | |
| Mitigated | 32.2 | 7.1 | −25.1 |
| Non-mitigated | 9.9 | 43.1 | +33.3 |
| Total | 42.1 | 48.1 | +8.2 |
| **Site 3. Skeppdalsström** | | | |
| Mitigated | 25.3 | 0 | −25.3 |
| Non-mitigated | 8.4 | 9.0 | +0.6 |
| Total | 33.7 | 9.0 | −24.7 |

**Note:**
Data were standardized to allow comparisons within and among sites; see text for further explanation.

recorded for all amphibian observations, both on the road, along fences and in the tunnels. For animals on tunnel photos, movement direction (in or out) was noted and the minimum net number through the tunnel was calculated. Significant numbers were found of four species (all except great crested newt) so we included data on all amphibians, and we pooled the data on all species in the analyses. Most amphibians found on or approaching the road were alive (ca 83%).

Field methodology and data output for all three sites is described in further detail in technical reports (in Swedish; *Ekologigruppen, 2004*; *Syde, 2008*; *Collinder, 2010*; *Calluna, 2012*; *Peterson, 2013a*, *2013b*; *Andersson & Lundberg, 2015*; *Helldin, 2015*; *Helldin, Olsson & Andersson, 2018*).

## Data treatment and analyses

We standardized the available data on amphibian counts on and near the roads, along fences and in tunnels to be able to compare, as far as possible, each site before and after mitigation and the mitigated road section with adjacent non-mitigated sections. We summarized the number of amphibians found on and near the road (including along temporary fences at site 2) per night (site 1) or evening (sites 2–3) and 50 m road interval, assuming that these data were collected with a similar effort and expertise over the road section searched, and with a similar effort and expertise before and after mitigation, within each site.

To be able to tentatively compare the performance of different tunnels at a site, we calculated the number through each tunnel per night (at site 1) or number of movements (in + out) and the net number through each tunnel per 24 h-period (at sites 2–3). To assess the number of amphibians successfully crossing a mitigated road section through the tunnels we summarized the net number through all tunnels at the site.
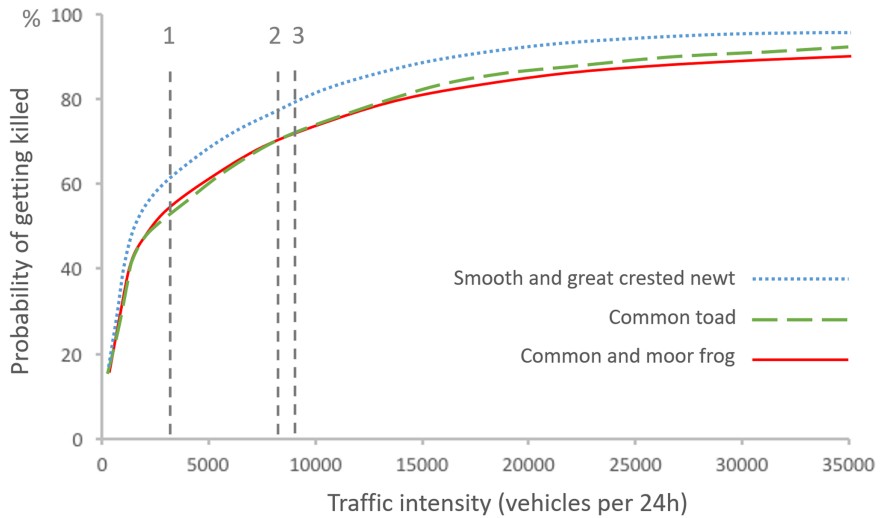

**Figure 4 Probability of getting killed for an individual of different amphibian species at different traffic intensities, as described by *Hels & Buchwald (2001)*.** The probability of getting killed is weighted by amphibian behavior (velocity and diurnal activity) and diurnal variation in traffic intensity, and assuming that amphibians are crossing perpendicular to the road. Traffic intensity of the three study sites are indicated by vertical dashed lines.

    To assess the number of amphibians killed and the number successfully crossing a non-mitigated road section, we used the information presented by *Hels & Buchwald (2001)* on the risk of getting killed for an amphibian on the road depending on average traffic intensity and species (Fig. 4). According to this relationship, a proportion of the amphibians attempting to cross a road should make it successfully to the other side even without any mitigation, i.e.:

$$x = 1 - y \tag{1}$$

where $x$ is the number of successful crossings and $y$ is the risk of getting killed on the road. Concomitantly, a number of amphibians found dead on the road should also represent a certain number that survived and managed to cross, following:

$$x = z\left(\frac{1}{y} - 1\right) \tag{2}$$

where $x$ is the number of successful crossings, $z$ is the number of amphibians found dead on the road, and $y$ is the risk of getting killed on the road. Based on average traffic intensity at site 1, we estimated that 62% of newts trying to cross the road surface would be killed by traffic (as in Fig. 4), and that each newt found dead represented 0.61 newt that had managed to cross (following Eq. (2)). Similarly, for site 2 we estimated a 70% risk of traffic mortality for toads (Fig. 4), with each toad found killed representing 0.43 toads that had crossed successfully (Eq. (2)) and each toad found along the temporary fence representing 0.30 toad that would have managed to cross the road, had the fence not been in place (Eq. (1)). Finally, for site 3 we assumed that on average 75% of amphibians trying to cross the road surface would get killed by traffic (an estimate based on 79% risk for newts, and 72% risk for toads and frogs; Fig. 4) and that each amphibian rescued

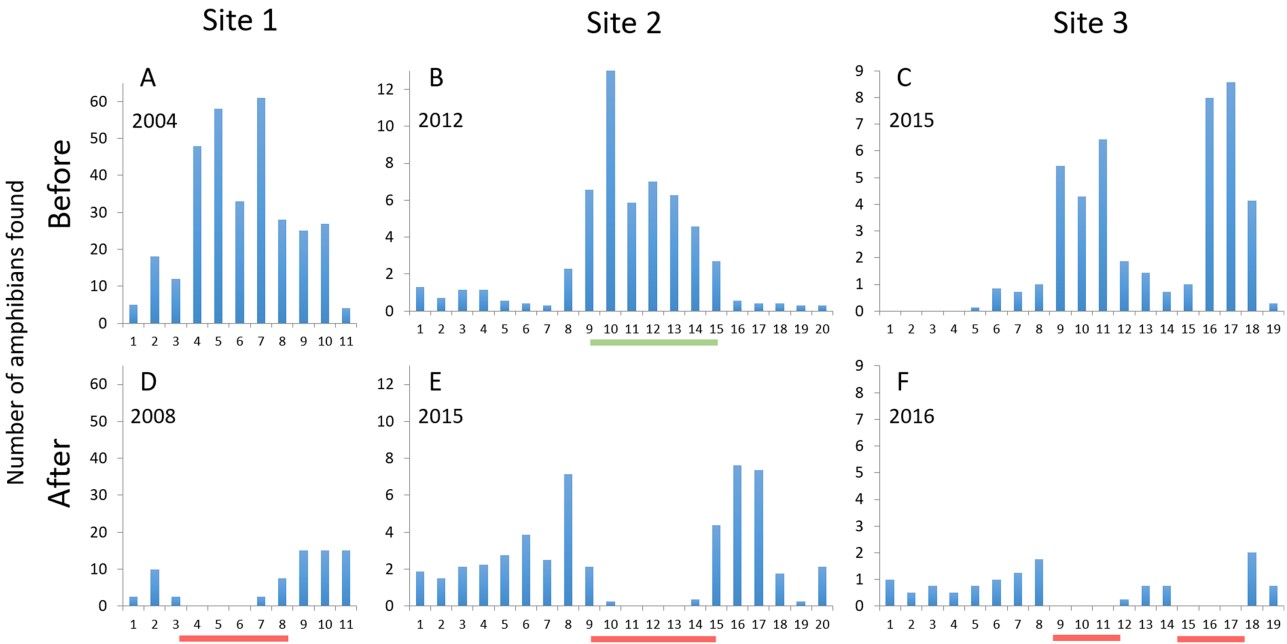

**Figure 5** **The number of amphibians found along the studied road sections, divided per evening or night and 50 m road interval starting from northwest.** Upper graphs (A–C) are before mitigation, lower graphs (D–F) are with mitigation in place. Site 1: Number of dead newts (smooth + great crested) found per night; site 2: Number of live and dead common toads found per night; site 3: Number of live and dead amphibians (four species) found per evening. Red lines below x-axes after mitigation denote the mitigated sections (permanent amphibian fencing), green line below x-axis at site 2 before mitigation denotes the temporary fenced section. Due to the difference in method, the data from counts along the temporary fence at site 2 cannot be directly compared to the other data from that site.

represented 0.25 amphibian that would have managed to cross the road, had the rescue not taken place (Eq. (1)).

## RESULTS

The number of amphibians found on or heading for the road, i.e., animals killed or at risk of being killed by car traffic, during spring migration decreased at mitigated road sections at all three sites (Fig. 5). The estimated number of individual amphibians saved by the mitigation measures ranged from 25 to >200 per night at the three sites (Table 3), corresponding to an 85–100% decrease in amphibians killed on the road along mitigated road sections. Outside mitigated sections, the changes from before to after mitigation installation were smaller and more variable; the number of amphibians on the road decreased by 33% at site 1, increased by over 300% at site 2, while there was virtually no change at site 3. At site 2, the number of amphibians on the road peaked just outside of the fence-ends (intervals 8 and 15–17; see Fig. 5). At sites 1 and 2, some individuals were found on the road just inside the fence-ends (east end at site 1, both ends at site 2; Fig. 5). No amphibians were found on a fenced road section >100 m from a fence-end.

The number of amphibians passing through the tunnels varied greatly between sites (3,000% difference; Table 4), generally in line with the numbers killed before mitigation, i.e., many more at site 1. The estimated number of amphibians successfully crossing the road increased at mitigated sections, ranging from 2 to 164 more individuals

**Table 4 Number of amphibian recordings in the tunnels, and the net number passing through per night or 24 h-period.**

**Site 1. Skårby (only newts, five nights during peak migration period)**

| Tunnel no. | S newt | GC newt | Both sp. | Net no./night |
|---|---|---|---|---|
| 1 | 473 | 145 | 618 | 123.6 |
| 2 | – | – | – | – |
| 3 | 21 | 28 | 49 | 9.8 |
| 4 | 612 | 90 | 702 | 140.4 |
| 5 | 111 | 5 | 116 | 23.2 |
| Sum | 1,217 | 268 | 1,485 | 297.0 |

**Site 2. Kyrksjölöten (only common toad, 14 significant migration days)**

| Tunnel no. | In | Out | Net no. | In + out/24 h | Net no./24 h |
|---|---|---|---|---|---|
| 1 | 871 | 397 | 474 | 90.6 | 33.9 |
| 2 | 545 | 216 | 329 | 54.4 | 23.5 |
| Sum | 1,416 | 613 | 803 | 144.9 | 57.4 |

**Site 3. Skeppdalsström (all amphibians, 7–11 days during peak migration period)**

| Tunnel no. | In | Out | Net no. | In + out/24 h | Net no./24 h |
|---|---|---|---|---|---|
| 1 (9 days) | 41 | 17 | 24 | 6.4 | 2.7 |
| 2 (11 days) | 258 | 254 | 4 | 46.5 | 0.4 |
| 3 (7 days) | 70 | 38 | 32 | 15.4 | 4.6 |
| 4 (7 days) | 20 | 0 | 20 | 2.9 | 2.9 |
| 5 | – | – | – | – | – |
| Sum | 389 | 309 | 80 | 71.2 | 10.5 |

**Note:**

For sites 2–3 (cameras) data are separated between animals moving into the tunnel (i.e., toward the breeding wetland) and those moving out (away from the wetland). At site 1 (traps), only animals moving toward the wetland could be counted, as net traps blocked the tunnels in the other direction. Tunnels that were not monitored are indicated by lack of data.

per night (Table 5), corresponding to a 25–340% increase compared to the situation before mitigation. In addition, the estimated number successfully crossing along non-mitigated sections differed before and after mitigation, and over the entire site (mitigated + non-mitigated road sections combined) the mitigation implementation resulted in 2–145 more individuals crossing the road per night (Table 5), or a 20–340% increase.

The number of amphibians passing through the tunnels also varied greatly among the tunnels at sites 1 and 3 (Table 4). Tunnel no. 2 at site 3 stood out by the large discrepancy between the high number of amphibians moving in and out of the tunnel entrance and the low net number passing through. This tunnel had a shallow pool in the northern (entrance) side, while the southern (exit) side was completely submerged due to a construction fault.

# DISCUSSION

The compiled results from the monitoring of amphibian passages at the three sites (Skårby, Kyrksjölöten, Skeppdalsström) indicate that the passages were effective in reducing

**Table 5 Estimated number of amphibians successfully crossing the road per night along the studied road sections before and after mitigation, separated between mitigated and adjacent non-mitigated sections.**

| Section | Before | After | Δ |
|---|---|---|---|
| **Site 1. Skårby** | | | |
| Mitigated | 139.1 | 303.1[a] | +164.0 |
| Non-mitigated | 55.5 | 36.6 | −18.9 |
| Total | 194.6 | 339.7 | +145.1 |
| **Site 2. Kyrksjölöten** | | | |
| Mitigated | 13.8 | 60.5[a] | +46.6 |
| Non-mitigated | 4.3 | 18.5 | +14.3 |
| Total | 18.1 | 79.0 | +60.9 |
| **Site 3. Skeppdalsström** | | | |
| Mitigated | 8.4 | 10.5[a] | +2.1 |
| Non-mitigated | 2.8 | 3.0 | +0.2 |
| Total | 11.2 | 13.5 | +2.3 |

**Notes:**
Data were standardized to allow comparisons within and among sites; see text for further explanation.
[a] Including the number passing through tunnels; see Table 4.

amphibian roadkill during spring migration, compared to the situation before mitigation measures were implemented. None or very few amphibians were found on the fenced road sections, where prior to mitigation amphibians had been killed in the hundreds or thousands each spring. These results are well in line with those from many other studies, showing significant reductions in amphibian roadkill after the construction of adequate road fences (*Meinig, 1989*; *Dodd, Barichivich & Smith, 2004*; *Jochimsen et al., 2004*; *Stenberg & Nyström, 2009*; *Malt, 2011*; *Matos et al., 2017, 2019*; *Hill et al., 2018*; *Jarvis, Hartup & Petrovan, 2019*).

However, the data from at least two of our sites suggested the presence of fence-end effects (*Huijser et al., 2016*) which may influence the overall reduction in amphibian roadkill. Peaks in numbers of amphibians on the road just outside fence-ends at site 2 suggest that some individuals following the fence by-passed the final portions of fencing, despite the angled design, and that part of the mortality was merely transferred from fenced to unfenced road sections. The increase in amphibians on the entire unfenced part of the road at site 2 may also be explained by individuals finding new migration routes when the previous ones have been occupied by fences, while tunnels are avoided or simply not encountered (though we also see several alternative explanations to that pattern; see below). Furthermore, at site 1 and site 2 some amphibians cut into the mitigated road section near the fence-ends. This may be an effect of animals moving diagonally over the road, not being strictly directional in their movements, or following the road along curbs or other minor structures into the fenced section. Nearer to the middle of the fenced sections, no amphibians were found on the road, and accordingly, in the central parts of the mitigated road sections the decrease in roadkilled amphibians was 100% at all three sites.

These fence-end effects, and the fact that many amphibians crossed and were killed on the road outside the fenced sections, imply that longer fences are likely to result in a larger reduction in roadkill (*Buck-Dobrick & Dobrick, 1989*; *Huijser et al., 2016*). While this notion may seem trivial, it has important implications for management (see below).

It is imperative that the effectiveness of amphibian passages in the form of under-road tunnels with associated guiding fences are not only assessed on the basis of the reduction in roadkill but also on the number of animals making it successfully to the other side of the road (*Jochimsen et al., 2004*; *Schmidt & Zumbach, 2008*). Previous studies have indicated that many amphibians reaching the fences do not find their way through the tunnels, either because the tunnels are too widely separated or the tunnels or guiding structures are inadequate, and as a consequence amphibians may return to the terrestrial habitats without breeding (*Allaback & Laabs, 2002*; *Jochimsen et al., 2004*; *Schmidt & Zumbach, 2008*; *Pagnucco, Paszkowski & Scrimgeour, 2012*; *Hedrick et al., 2019*). Several European studies have reported the overall rates of individual toads or newts using tunnels ranging from 3% to 98% of those encountering the guiding fences (*Brehm, 1989*; *Buck-Dobrick & Dobrick, 1989*; *Langton, 1989*; *Meinig, 1989*; *Zuiderwijk, 1989*; *Mechura et al., 2012*; *Matos et al., 2017, 2019*; *Ottburg & Van Der Grift, 2019*; *Jarvis, Hartup & Petrovan, 2019*).

The results from our three sites indicated that the mitigation schemes likely reduced the barrier effects of the roads. We assumed that even without mitigation in place, a certain proportion of amphibians manage to cross a road without getting killed by traffic, that most amphibians survive where the traffic intensity is very low, but that the proportion surviving decreases exponentially with increasing traffic (*Hels & Buchwald, 2001*; *Gibbs & Shriver, 2005*; *Jacobson et al., 2016*). Importantly, however, on all three sites studied, the number of individuals passing through the tunnels in spring exceeded the number estimated to have crossed the road surface successfully over the mitigated section before the mitigation was in place.

Several factors in the technical construction of amphibian passages may affect their effectiveness: width, shape and length of tunnels, distance between tunnels, height and shape of guiding barriers, substrate in tunnels and along barriers, construction material, moisture, vegetation and drainage in and around the passages, special features such as cover objects, guiding structures at entrances and slotted tops (reviews in *Jochimsen et al., 2004*; *Hamer, Langton & Lesbarrères, 2015*; *Jackson, Smith & Gunson, 2015*). Our data did not allow a systematic analysis of how these factors relate to the passage effectiveness. With the information at hand, we can only speculate about the differences observed. At site 1, many newts were carried through the tunnels by the water running in direction towards the wetland, and at site 3, standing water in one of the tunnels appeared to attract many amphibians to the tunnel entrance but blocked the tunnel for actual crossings. Shallow standing or running water in and around tunnels can attract amphibians and help them finding their way through (*Rosell et al., 1997*; *Eriksson, Sjölund & Andren, 2000*; *Jochimsen et al., 2004*; *Schmidt & Zumbach, 2008*, *Jarvis, Hartup & Petrovan, 2019*), but high water levels make tunnels impassable (*Buck-Dobrick & Dobrick, 1989*; *Rosell et al., 1997*; *Jochimsen et al., 2004*). Water levels may thus have a significant, but complex, impact on amphibian passage effectiveness. Additionally, the water and

soil inside and adjacent to amphibian tunnels can suffer high pollution levels from road surface contaminants including salt used for deicing roads as well as various metals and other substances (*White, Mayes & Petrovan, 2017*). At site 2, both the tunnels and the distance between them were longer than at the other sites, which may explain a bypass effect, i.e., peaks in animals on the road just outside fence-ends. Previous studies suggest that long tunnels and long fences without tunnels make amphibians give up and turn back (*Zuiderwijk, 1989*; *Jochimsen et al., 2004*; *Jackson, Smith & Gunson, 2015*; *Hill et al., 2018*; *Ottburg & Van Der Grift, 2019*; *Matos et al., 2019*); these individuals may eventually try crossing the road on another spot. There were substantial movements in and out of the tunnels at this site, which may also indicate that animals hesitated to pass through. However, the total numbers actually crossing through the tunnels were broadly similar to the estimated number killed or crossing the fenced section before mitigation (58.8/24 h vs. 32.1 + 13.8 = 45.9/night).

There are several plausible explanations for the changes in the number of amphibians on the road outside mitigated sections (most pronounced at sites 1 and 2), other than the potential bypass effect described above. The most obvious is that the field effort at some sites and time periods was insufficient (three nights or less for data collection) and the data therefore were influenced by random events. Another is that the fieldwork methods were in fact not similar enough with regard to how the basic method was applied in practice to allow the data standardization and comparisons. The changes observed may also depend on annual differences in population numbers or temporal migration patterns. In this case, the effect sizes on mitigated sections can be adjusted according to the changes on non-mitigated sections. However, it is important to note that the non-mitigated sections studied were not true controls (comparators), as they may have been affected by the mitigation measure (the intervention).

The standardization of data required a number of assumptions and simplifications that may have introduced errors. We adopted an approach where we tried finding the unifying patterns in studies of amphibian passages conducted with slightly different aims, budgets, staffing and time frames. Despite these limitations, which are unfortunately common in applied conservation projects, we believe that the general picture given by these studies, before vs. after mitigation and along vs. outside the mitigated road section, contributes significantly to the knowledge of how amphibian passages at roads can reduce roadkill and barrier effects on amphibians during spring migration.

## CONCLUSIONS

There is scant evidence in literature that amphibian passages lead to long-term conservation of amphibian populations (*Beebee, 2013*; *Smith, Meredith & Sutherland, 2018*; *Jarvis, Hartup & Petrovan, 2019*), and for our three sites it is difficult to be certain to what degree the observed reductions in roadkill and barrier effect will have a significant and long-lasting effect on the population level. However, the estimated number of newts saved by the mitigation system (>200 individuals per peak migration night) and the number of newts crossing through the tunnels (ca 180 per peak migration night) at site 1 (Skårby) are each in the same order of magnitude as the total estimated number

of breeding newts at the site (2,000–2,300 individuals, assuming that there are around 10 peak migration nights per season; *Peterson & Collinder, 2006*).

By contrast, the low number of amphibians successfully crossing through the tunnels at site 3 (Skeppdalsström)—ca 10 individuals per night, an increase with only two per night compared to what may have crossed the road successfully without any mitigation—may appear discouraging. The reduction in the number killed (some 25 per peak migration night) sums up to nowhere near the total estimated number of amphibians breeding at the site (ca 1,300 individuals; *Andersson & Lundberg, 2015*). The results from site 2 (Kyrksjölöten) indicate that many more toads manage to cross the road alive using the tunnels compared to before mitigation, but these results cannot be put in relation to any estimated population size, and the conclusion regarding the benefit to conservation is confused by the possible bypass effects (see above).

It is important to point out that there should be a minimal level of road traffic where amphibian passages of the kind described here need to be considered, as implied by the relationship between traffic intensity and risk of getting killed described by *Hels & Buchwald (2001)*, Fig. 4. On roads with low traffic many amphibians are likely to cross the road without getting killed, and an amphibian passage with fences that hinders some of these movements may lead to a decrease in the number of successful crossings, and accordingly cause more harm than good (*Jaeger & Fahrig, 2004*; *Jochimsen et al., 2004*; *Schmidt & Zumbach, 2008*; *Pagnucco, Paszkowski & Scrimgeour, 2012*). The cut-off point depends on the combination of traffic intensity and effectiveness of passages.

All data treatment in our work relies heavily on *Hels & Buchwald's (2001)* risk model for amphibians. While that study was well conducted, the results were based on few species and limited observations, and empirical tests of the model prediction are still rare (*Gibbs & Shriver, 2005*). Given the need for road managers to know under what circumstances the construction of amphibian passages is motivated, and when not, we strongly recommend further study of the relation between road characteristics (traffic, width, etc.) and the roadkill risk for amphibians when attempting to cross.

At all three sites the mitigation was restricted solely to the most critical road sections (see Fig. 5), despite recommendations in ecological assessments from all sites to include also contiguous sections (*Collinder, 2007*; *Helldin, 2015*; *Lundberg, 2015*). Our results suggest that mitigation (guiding fences and additional tunnels) extending at least some 100 m outside of the most critical road section could minimize fence-end effects and further improve the passage effectiveness.

An alternative approach to decrease fence-end effect could be to fortify fence-ends, for example, by modifying the angles or extending fences perpendicularly from the road, compared to what was done at our sites 2 and 3 (Figs. 3D and 3F). Amphibians could potentially be helped in finding and entering tunnels with relatively simple means by installing guiding structures at the tunnel entrances where these are not already in place (site 3). However, it is unclear to what degree such adaptations would improve the effectiveness of existing passages.

Amphibian passages tend to be costly, not least when constructed on existing roads, and it is therefore crucial for road managers to know where passages may be critical

for amphibian conservation and how passages can best be designed. To build up the knowledge of amphibian passages at roads, the reduction in roadkill and barrier effects should be monitored when new amphibian passages are constructed, or when existing passages are adapted (*Hamer, Langton & Lesbarrères, 2015*; *Helldin, 2017*). The monitoring should use comparable methods before and after mitigation, include the quantification of amphibians killed and amphibians successfully crossing, over a long enough road section to cover bypass effects. Quality data should be secured by a field effort spanning multiple years before and after mitigation, and multiple times each year. Results from such studies could be combined in regional and global analyses (e.g., meta-analyses) to explore differences between construction types and trade-offs between the economic investment and expected effect size (cost-efficiency), thereby helping to point out where passages along existing roads are warranted.

Finally, it is important to note that our results only focused on adult breeding migrations in spring, without including the summer and autumn migrations of juveniles away from the breeding ponds. Recent population models indicate that the survival rate of post-metamorphic juveniles is of fundamental importance for the persistence of amphibian populations (*Schmidt & Zumbach, 2008*; *Petrovan & Schmidt, 2019*). Adults and juveniles using the passages later in the season, when leaving the breeding areas, may experience dryer tunnels or even water counterflow. Juvenile amphibians may be particularly sensitive to the design of underpasses and associated barrier fences (*Schmidt & Zumbach, 2008*) given their higher desiccation risk. However, due to their very small size and unpredictable migration timing, juveniles remain very rarely quantified in terms of both road mortality impacts and usage of mitigation systems, despite their crucial role in population dynamics (*Petrovan & Schmidt, 2019*). Future studies should prioritize incorporating juveniles in mitigation assessments.

## ACKNOWLEDGEMENTS

We thank the following people for collecting and processing field data used in the study: Petter Andersson, Per Collinder, Abel Gonzales, Michael Hartup, Mova Hebert, Anna Koffman, Johanna Lundberg, Terese Olsson, Torbjörn Peterson, Anna Seffel, Lisa Sigg, Nina Syde, Claes Vernerback and Mikael Åsberg. We are particularly grateful to Per Collinder for digging out old Skårby reports, and to all the volunteers at Djurens Ö Wildlife Rescue for backing up the field work at Skeppdalsström. We thank Lars Nilsson, Edgar van der Grift, Brandon Hedrick and Richard Griffiths for reading and commenting on previous versions of this paper.

### Funding

The analysis and writing of this article was conducted with research funding from the Swedish Transport Administration (project TRIEKOL, TRV2016/50073). The funders had no role in study design, data collection and analysis, decision to publish, or preparation of the manuscript.

### Grant Disclosures

The following grant information was disclosed by the authors:
Swedish Transport Administration: TRIEKOL, TRV2016/50073.

### Competing Interests

Silviu O. Petrovan is non-paid trustee (board of directors) of Froglife, a charity (non-profit) NGO.

### Author Contributions

- Jan Olof Helldin conceived and designed the experiments, performed the experiments, analyzed the data, contributed reagents/materials/analysis tools, prepared figures and/or tables, authored or reviewed drafts of the paper, approved the final draft.
- Silviu O. Petrovan analyzed the data, contributed reagents/materials/analysis tools, authored or reviewed drafts of the paper, approved the final draft.

### Data Availability

Raw data are presented in technical reports at the following URLs:

http://media.triekol.se/2019/03/Ekologigruppen-2004-Utredning-Skarby.pdf

http://media.triekol.se/2019/03/Syde-2008-Groddjursinventering-vid-konfliktpunkter-i-Stockholms-lan.pdf

http://media.triekol.se/2019/03/Collinder-2010-Groddjursinventering-2010-vid-konfliktpunkter-i-Stockholms-lan.pdf

http://miljobarometern.stockholm.se/content/docs/tema/natur/Groddjur/Groddjursrapport_Spangavagen121219_Lagupplost.pdf

http://media.triekol.se/2019/03/Peterson-2013-Utvardering-av-grodtunnlar-i-Skarbydammen.pdf

http://media.triekol.se/2019/03/Andersson-Lundberg-2015-Groddjursinventering-och-flytt-vid-vag-222.pdf

http://miljobarometern.stockholm.se/content/docs/tema/natur/Groddjur/Groddjurstunnlar%20Sp%C3%A5ngav%C3%A4gen%20Rapport%202015-10-02.pdf

https://trafikverket.ineko.se/Files/sv-SE/57352/Ineko.Product.RelatedFiles/2018_232_uppfoljning_och_utvardering_av_groddjursatgarder_vid_skeppdalsstrom.pdf

Overview of standardized raw data used in the analyses are available in Article S2.

### Supplemental Information

Supplemental information for this article can be found online at http://dx.doi.org/10.7717/peerj.7518#supplemental-information.

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
