# Peer review of "Effectiveness of small road tunnels and fences in reducing amphibian roadkill and barrier effects at retrofitted roads in Sweden"

_PeerJ, doi:10.7717/peerj.7518_

## Round 0.1 · original submission · Major Revisions

Both reviewers have made some important comments that will improve your manuscript. I encourage you to consider those comments and modify the manuscript accordingly.

·

Basic reporting

No comment

Experimental design

No comment

Validity of the findings

No comment

Additional comments

Helldin and Petrovan Review: Effectiveness of small road tunnels and fences in reducing amphibian roadkill and barrier effects; case studies of retrofitted roads in Sweden

The work done by Helldin and Petrovan presents an important contribution to our understanding of the use of tunnels and drift fencing on amphibian roadkill mitigation. I think the study is effective and the conclusions are justly drawn. However, there are several comments that I think would make the paper even more effective and increase its citability. I would be happy to review a revision of this paper in the future.

Brandon Hedrick

Major comments:

1) Please include the supplemental information on each site in the main manuscript. Considering the PeerJ has a digital format, I wouldn’t want this information to be lost in the supplement. When I read the MS, I wanted a more thorough explanation of the site differences earlier on and I think this would remedy the problem.
2) I would strongly consider combining figures 2–4 into a multiple panel figure with the images of the tunnels in the supplement. While table 1 lists the differences between tunnels, a visual would really help show the differences.
3) I was a little confused until I looked at Table 2 what you meant by ‘before mitigation’ and ‘after mitigation’. There isn’t any mention of the years in the actual text and it reads like the ‘before mitigation’ and ‘after mitigation’ were several weeks apart.
4) There are a few instances where methods are not adequately described. I am not sure how the data from lines 300–307 was generated. Please add this to the methods. More information on how the Hels and Buchwald equation was applied at each site (what the variables were) could be presented more clearly, perhaps in a table. I was also a little confused as to how the data were standardized. Please add in more details to the methods to remedy this.

Line by Line:

Line 25: Please spell out versus for the abstract

Line 26: consider ‘In mitigated road sections, the estimated number..’. I had a little trouble following as written.

Line 37: change to ‘by road mortality, barrier effects of roads, and traffic’

Line 42: ‘in attempt’

Line 40-42: Is there any evidence of road avoidance in amphibians? I have seen many salamanders and frogs walk across roads blissfully unaware of traffic. Can you add a citation?

Line 47: Please add the recent Hedrick et al. (2019) pub here: https://www.biorxiv.org/content/10.1101/569426v1

Line 65: Change ‘where’ to ‘were’

Line 105: Since there is no standard for monitoring tunnels, I think the supplement would be better moved back into the main paper. I would hate for this important information to be lost in the supplement. The supplement will need to be edited some in order to pull it into the main manuscript.

Line 114: I’m not sure why site 2 was so different having read the supplement. Is it just that there were temporary fences before mitigation?

Line 130: ‘..information presented by Hels and Buchwald..”

Line 142: For site 3, did you use the same speeds for the two newt species and for the two frog species in the Hels and Buchwald equation? I think more explanation is needed here.

Line 162: What do you mean when you say ‘largely following the number that was killed before mitigation’?

Line 179: Roadkill should not be a verb throughout
Line 192: The fence end effects can be curbed by putting a loop at the end of the fences to reorient the amphibians. See page 9 here: https://files.ontario.ca/environment-and-energy/species-at-risk/mnr_sar_tx_rptl_amp_fnc_en.pdf
Line 205–216: Add in the recent Hedrick et al. (2019) paper here as well. That study focused on this issue in particular.
Line 221: Gibbs and Shriver 2005 can also be cited here. https://link.springer.com/article/10.1007/s11273-004-7522-9
Line 241: ‘…have a significant, but complex..’

Line 244: Longer than at the other sites right?

Line 262: Change ‘not unaffected’ to ‘affected’

Line 283: Change to ‘In contrast’

Line 286: I’m not sure what the sentence starting with ‘neither’ means. Reword.

Line 300–307: I’m not sure how this data was generated. Can you add this to the methods or delete?

Line 308: Gibbs and Shriver (2005) above used the Hels and Buchwald equation for a different species and confirmed its use.

Line 314: I think another very important point here is to identify the hotspot. It is not uncommon for tunnels to be placed near a hotspot, but not on it. In this case, the tunnels are not used at all.

Line 320: See the comment about the fence loop above.

Line 333: I completely agree! Too often tunnels are placed based on scant data and their impact is not assessed.

Line 350: Very good point about the juveniles. I wish they were easier to monitor. I have also always ignored this important factor in tunnel studies.

Table 2 makes it look like some sites were not measured before or after mitigation. Can you add in a row that says something like ‘manual monitoring’ to show how they were monitored?

·

Basic reporting

This is very good. The whole paper is very well written, explains the problem that is to be addressed clearly, and places the research firmly within a body of existing literature. The structure conforms to PeerJ standards and the figures and tables are sensibly structured and clear. I enjoyed reading it and found the results interesting, even though the different methods used at the different sites took some digesting.

Experimental design

The question posed is a sound and relevant one – how effective are amphibian tunnels in reducing road mortalities? The rationale for tackling this is also very sound, with two levels of controls – before and after the intervention was implemented (repeated measures) and mitigated vs unmitigated sites. The three sites provide some degree of replication which is also good. What is problematical, however, is that the study is collating information from three projects that were designed in rather different ways in terms of logistics, species, timings, tunnel/fencing design etc. This is very common in ‘real world’ conservation practice where the original goal was to perform an intervention rather than perform a rigorous experiment. I am therefore very sympathetic to the challenges faced in delivering a rigorous experimental outcome given these constraints. The fact remains, however, that the inherent inconsistencies in how these three projects were carried out present some significant issues in terms of rigorous analysis. To overcome this, the authors have valiantly attempted to standardize the data from the three studies as far as they can so comparisons can be made. Nevertheless, this means that the analysis is entirely descriptive without any statistical testing. Personally, I can live with this given the design, but it does mean that there are numerous imponderables that make direct comparisons between before and after, and between mitigated and unmitigated sites, rather difficult. The authors acknowledge these issues and discuss them in a very articulate way in the paper. However, it does leave the reader with a feeling that there may be some doubt about the veracity of the findings. I am not sure how to address this, unless some sort of test (probably non-parametric) of association can be performed between (for example) ‘Change in number of deaths on road’ and ‘before and after mitigation’ and ‘mitigation vs non-mitigation’. Whether or not the paper warrants publication without some formal hypothesis test may therefore be an editorial matter.

Validity of the findings

There is compelling descriptive analysis (based on the standardization procedure) that the mitigation interventions reduced road mortality of amphibians. However, as described above, these descriptive analyses do make a range of assumptions about the data (e.g. percentage mortality can be modelled from an estimate of traffic intensity based from a different study), and there are range of possible confounding factors that might have influenced the results (e.g. seasonal changes in behaviour, natural population fluctuations between years, differences in survey effort between sites etc). The Discussion is sensibly cautious in that it flags such issues, and I suspect the authors are correct in their interpretation of the data. But it does raise the question of whether the study is technically and analytically sound enough to meet PeerJ editorial requirements. Again, working in practical conservation myself I understand the constraints and can live with this, but ultimately I think this issue is an editorial call.

Additional comments

Line 63: I suggest ‘three independent sites’ rather than ‘hot-spots’ is more informative here.
Line 65: ‘were’ rather than ‘where’.
Lines 109-113: Some information is given in the supplementary information about the number of personnel involved in the surveys. How did the standardization procedure control for variation in personnel and expertise?
Lines 113-115: I wonder if site 2 is too anomalous to include in the study. The actual number of amphibians that appeared to be killed actually increased after the mitigation (Table 3). The reasons for this are explained, but it does rather throw the story.
Lines 126-127: Sorry, I can’t follow the rationale about how the number killed on the road could represent a certain number that survived.
Lines 255-257: This is an honest and pragmatic admission, but it does raise some doubt about the reliability of the interpretation of the increases/decreases.
Lines 261-265: Ditto last comment.
Lines 267-270: I suspect the authors are correct in this assertion, but a purist might argue otherwise.
Lines 281-282: I think the assertion that decrease in breeding adult deaths = increase in adult survival = benefit to conservation makes some assumptions. Amphibians have complex life cycles and density dependence may operate at different stages. It is therefore plausible that reduced survival in the adult stage could relax competition in tadpole/juvenile stage and thereby compensate for the mortality. As the paper deals with ‘deaths on roads’ rather than ‘survival’ (sensu stricto), it may be safer just to remove this sentence unless there is supporting literature.
Lines 290-291: A possible reason to remove site 2 from the analysis?
Lines 302-306: I quite like the idea of calculating thresholds like this. Any idea how these theoretical estimates related to traffic at these sites in reality?
Lines 306-307: Another reason to dump site 2 from the analysis?!
Lines 326-333: Without wanting to divert the paper down an economic line, this interesting point does raise the logical question of when does it become more cost-effective to build new habitat on one side of the road rather than build a tunnel?
Lines 333-338: Following on from the last point (and with the same qualifier), all these points are very good ones with which I entirely agree. However, the stark reality is who pays for this research to be done? I would envisage road building companies, government, research funders and conservation agencies all passing the buck to each other. This is a very common problem with no easy way forward.
References: Check these carefully. A few have capitalized first letters of words in article title.
Table 1: ‘Height’ not ‘Hight’
Figure 5: I assume that y-axis plots ‘number of evenings/nights’? May be wise to explicitly label it.

---

## Round 0.2 · Minor Revisions

A reviewer of the original manuscript has complimented you on your revisions, but has a few minor changes/suggestions that may help with clarity.

·

Basic reporting

no comment

Experimental design

no comment

Validity of the findings

no comment

Additional comments

Helldin and Petrovan Review #2

The authors have responded to all of my previous comments adequately and I feel that the paper is quite strong now. I have suggested several additional minor changes to correct spelling and grammatical errors that should be incorporated before publication. However, I do not think that it is necessary for me to review the manuscript again before publication. Great work! I am excited to see that this is getting out.

Brandon Hedrick


Minor Comments:

Line 58: I originally read road tunnels as tunnels for cars. Can you change to ‘road culverts’ or ‘under-road tunnels’ to clarify?

Line 105: Change ‘Swedens’ to ‘Sweden’s’

Line 113: Change ‘trapping in tunnels was conducted in during’

Line 133: Change ‘significant’ to ‘large’ or something like that throughout. Otherwise, define what constitutes a ‘significant migration night’ statistically.

Line 139: ‘therefore’ spelling

Line 123, 149, 179: Please delete the last sentence at all sites and move it to the end of the section before site descriptions. It is repetitive as is.

Line 202: Change to ‘should also represent’

Line 208: This sentence is a little awkward. Please change

Line 311: Change to: ‘…may thus have a significant, but…’

Line 321, 341, 348: I would prefer a word other than significant. Significant is associated with a p-value. Maybe substantial would fit better.

Line 371: Add (2001) to Hels and Buchwald here

Line 372: ‘..while that study was well conducted, the results were based..’

Line 382: I can’t find the citation, but there have been suggestions in mitigation plans of having a loop at the end of fences to funnel amphibians back into the tunnel system. You might discuss that here. It looks like some of the fences you used incorporated this to varying degrees (Fig 3D, F)

Line 483: Need new paragraph

Table 4: I find it really interesting that you had such variable tunnel success at sites 2 and 3. Less than half of the critters entering the tunnels at site 2 actually exited. However, 80% used the tunnels successfully at site 3. I wonder why.

---

## Round 0.3 · accepted · Accept

Thank you for your efforts in revising your manuscript. I appreciate your attention to detail.